# The Limitations of Adversarial Training and the Blind-Spot Attack

**Huan Zhang**[1]*, **Hongge Chen**[2]*, **Zhao Song**[3], **Duane Boning**[2], **Inderjit Dhillon**[3], **Cho-Jui Hsieh**[1]

[1]UCLA, Los Angeles, CA 90095
[2]MIT, Cambridge, MA 02139
[3]UT Austin, Austin, TX 78712

`huan@huan-zhang.com, chenhg@mit.edu, zhaos@utexas.edu`
`boning@mtl.mit.edu, inderjit@cs.utexas.edu, chohsieh@cs.ucla.edu`
*Huan Zhang and Hongge Chen contributed equally to this work.

## Abstract

The adversarial training procedure proposed by Madry et al. (2018) is one of the most effective methods to defend against adversarial examples in deep neural networks (DNNs). In our paper, we shed some lights on the practicality and the hardness of adversarial training by showing that the effectiveness (robustness on test set) of adversarial training has a strong correlation with the distance between a test point and the manifold of training data embedded by the network. Test examples that are relatively far away from this manifold are more likely to be vulnerable to adversarial attacks. Consequentially, an adversarial training based defense is susceptible to a new class of attacks, the *"blind-spot attack"*, where the input images reside in "blind-spots" (low density regions) of the empirical distribution of training data but is still on the ground-truth data manifold. For MNIST, we found that these blind-spots can be easily found by simply scaling and shifting image pixel values. Most importantly, for large datasets with high dimensional and complex data manifold (CIFAR, ImageNet, etc), the existence of blind-spots in adversarial training makes defending on any valid test examples difficult due to the curse of dimensionality and the scarcity of training data. Additionally, we find that blind-spots also exist on provable defenses including (Kolter & Wong, 2018) and (Sinha et al., 2018) because these trainable robustness certificates can only be practically optimized on a limited set of training data.

## 1 Introduction

Since the discovery of adversarial examples in deep neural networks (DNNs) (Szegedy et al., 2013), adversarial training under the robustness optimization framework (Madry et al., 2018; Sinha et al., 2018) has become one of the most effective methods to defend against adversarial examples. A recent study by Athalye et al. (2018) showed that adversarial training does not rely on obfuscated gradients and delivers promising results for defending adversarial examples on small datasets. Adversarial training approximately solves the following min-max optimization problem:

$$\min_{\theta} \mathbb{E}_{(x,y) \in \mathcal{X}} \left[ \max_{\delta \in S} L(x + \delta; y; \theta) \right], \tag{1}$$

where $\mathcal{X}$ is the set of training data, $L$ is the loss function, $\theta$ is the parameter of the network, and $S$ is usually a norm constrained $\ell_p$ ball centered at 0. Madry et al. (2018) propose to use projected gradient descent (PGD) to approximately solve the maximization problem within $S = \{\delta \mid \|\delta\|_\infty \leq \epsilon\}$, where $\epsilon = 0.3$ for MNIST dataset on a 0-1 pixel scale, and $\epsilon = 8$ for CIFAR-10 dataset on a 0-255 pixel scale. This approach achieves impressive defending results on the MNIST test set: so far the best available white-box attacks by Zheng et al. (2018) can only decrease the test accuracy from approximately 98% to 88%[1]. However, on CIFAR-10 dataset, a simple 20-step PGD can decrease the test accuracy from 87% to less than 50%[2].

---

[1] https://github.com/MadryLab/mnist_challenge    [2] https://github.com/MadryLab/cifar10_challenge

The effectiveness of adversarial training is measured by the robustness on the test set. However, the adversarial training process itself is done on the training set. Suppose we can optimize (1) perfectly, then certified robustness may be obtained on those training data points. However, if the empirical distribution of training dataset differs from the true data distribution, a test point drawn from the true data distribution might lie in a low probability region in the empirical distribution of training dataset and is not "covered" by the adversarial training procedure. For datasets that are relatively simple and have low intrinsic dimensions (MNIST, Fashion MNIST, etc), we can obtain enough training examples to make sure adversarial training covers most part of the data distribution. For high dimensional datasets (CIFAR, ImageNet), adversarial training have been shown difficult (Kurakin et al., 2016; Tramèr et al., 2018) and only limited success was obtained.

A recent attack proposed by Song et al. (2018) shows that adversarial training can be defeated when the input image is produced by a generative model (for example, a generative adversarial network) rather than selected directly from the test examples. The generated images are well recognized by humans and thus valid images in the ground-truth data distribution. In our interpretation, this attack effective finds the "blind-spots" in the input space that the training data do not well cover.

For higher dimensional datasets, we hypothesize that many test images already fall into these blind-spots of training data and thus adversarial training only obtains a moderate level of robustness. It is interesting to see that for those test images that adversarial training fails to defend, if their distances (in some metrics) to the training dataset are indeed larger. In our paper, we try to explain the success of robust optimization based adversarial training and show the limitations of this approach when the test points are slightly off the empirical distribution of training data. Our main contributions are:

- We show that on the original set of test images, the effectiveness of adversarial training is highly correlated with the distance (in some distance metrics) from the test image to the manifold of training images. For MNIST and Fashion MNIST datasets, most test images are close to the training data and very good robustness is observed on these points. For CIFAR, there is a clear trend that the adversarially trained network gradually loses its robustness property when the test images are further away from training data.
- We identify a new class of attacks, "*blind-spot attacks*", where the input image resides in a "blind-spot" of the empirical distribution of training data (far enough from any training examples in some embedding space) but is still in the ground-truth data distribution (well recognized by humans and *correctly classified* by the model). Adversarial training cannot provide good robustness on these blind-spots and their adversarial examples have small distortions.
- We show that blind-spots can be easily found on a few strong defense models including Madry et al. (2018), Kolter & Wong (2018) and Sinha et al. (2018). We propose a few simple transformations (slightly changing contrast and background), that do not noticeably affect the accuracy of adversarially trained MNIST and Fashion MNIST models, but these models become vulnerable to adversarial attacks on these sets of transformed input images. These transformations effectively move the test images slightly out of the manifold of training images, which does not affect generalization but poses a challenge for robust learning.

Our results imply that current adversarial training procedures cannot scale to datasets with a large (intrinsic) dimension, where any practical amount of training data cannot cover all the blind-spots. This explains the limited success for applying adversarial training on ImageNet dataset, where many test images can be sufficiently far away from the empirical distribution of training dataset.

## 2 RELATED WORKS

### 2.1 DEFENDING AGAINST ADVERSARIAL EXAMPLES

Adversarial examples in DNNs have brought great threats to the deep learning-based AI applications such as autonomous driving and face recognition. Therefore, defending against adversarial examples is an urgent task before we can safely deploy deep learning models to a wider range of applications. Following the emergence of adversarial examples, various defense methods have been proposed, such as defensive distillation by Papernot et al. (2016) and feature squeezing by Xu et al. (2017). Some of these defense methods have been proven vulnerable or ineffective under strong attack methods such as C&W in Carlini & Wagner (2017). Another category of recent defense methods is based on gradient masking or obfuscated gradient (Buckman et al. (2018); Ma et al.

(2018); Guo et al. (2017); Song et al. (2017); Samangouei et al. (2018)), but these methods are also successfully evaded by the stronger BPDA attack (Athalye et al. (2018)). Randomization in DNNs (Dhillon et al., 2018; Xie et al., 2017; Liu et al., 2018) is also used to reduce the success rate of adversarial attacks, however, it usually incurs additional computational costs and still cannot fully defend against an adaptive attacker (Athalye et al., 2018; Athalye & Sutskever, 2017).

An effective defense method is adversarial training, which trains the model with adversarial examples freshly generated during the entire training process. First introduced by Goodfellow et al., adversarial training demonstrates the state-of-the-art defending performance. Madry et al. (2018) formulated the adversarial training procedure into a min-max robust optimization problem and has achieved state-of-the-art defending performance on MNIST and CIFAR datasets. Several attacks have been proposed to attack the model release by Madry et al. (2018). On the MNIST testset, so far the best attack by Zheng et al. (2018) can only reduce the test accuracy from 98% to 88%. Analysis by Athalye et al. (2018) shows that this adversarial training framework does not rely on obfuscated gradient and truly increases model robustness; gradient based attacks with random starts can only achieve less than 10% success rate with given distortion constraints and are unable to penetrate this defense. On the other hand, attacking adversarial training using generative models have also been investigated; both Xiao et al. (2018) and Song et al. (2018) propose to use GANs to produce adversarial examples in black-box and white-box settings, respectively.

Finally, a few certified defense methods (Raghunathan et al., 2018; Sinha et al., 2018; Kolter & Wong, 2018) were proposed, which are able to provably increase model robustness. Besides adversarial training, in our paper we also consider several certified defenses which can achieve relatively good performance (i.e., test accuracy on natural images does not drop significantly and training is computationally feasible), and can be applied to medium-sized networks with multiple layers. Notably, Sinha et al. (2018) analyzes adversarial training using distributional robust optimization techniques. Kolter & Wong (2018) and Wong et al. (2018) proposed a robustness certificate based on the dual of a convex relaxation for ReLU networks, and used it for training to provably increase robustness. During training, certified defense methods can provably guarantee that the model is robust on training examples; however, on unseen test examples a non-vacuous robustness generalization guarantee is hard to obtain.

## 2.2 ANALYZING ADVERSARIAL EXAMPLES

Along with the attack-defense arms race, some insightful findings have been discovered to understand the natural of adversarial examples, both theoretically and experimentally. Schmidt et al. (2018) show that even for a simple data distribution of two class-conditional Gaussians, robust generalization requires significantly larger number of samples than standard generalization. Cullina et al. (2018) extend the well-known PAC learning theory to the case with adversaries, and derive the adversarial VC-dimension which can be either larger or smaller than the standard VC-dimension. Bubeck et al. (2018b) conjecture that a robust classifier can be computationally intractable to find, and give a proof for the computation hardness under statistical query (SQ) model. Recently, Bubeck et al. (2018a) prove a computational hardness result under a standard cryptographic assumption. Additionally, finding the safe area approximately is computationally hard according to Katz et al. (2017) and Weng et al. (2018). Mahloujifar et al. (2018) explain the prevalence of adversarial examples by making a connection to the "concentration of measure" phenomenon in metric measure spaces. Su et al. (2018) conduct large scale experiments on ImageNet and find a negative correlation between robustness and accuracy. Tsipras et al. (2019) discover that data examples consist of robust and non-robust features and adversarial training tends to find robust features that have strongly-correlations with the labels.

Both adversarial training and certified defenses significantly improve robustness on training data, but it is still unknown if the trained model has good *robust generalization* property. Typically, we evaluate the robustness of a model by computing an upper bound of error on the test set; specifically, given a norm bounded distortion $\epsilon$, we verify if each image in test set has a robustness certificate (Zhang et al., 2018; Dvijotham et al., 2018; Singh et al., 2018). There might exist test images that are still within the capability of standard generalization (i.e., correctly classified by DNNs with high confidence, and well recognized by humans), but behaves badly in robust generalization (i.e., adversarial examples can be easily found with small distortions). Our paper complements those existing findings by showing the strong correlation between the effectiveness of adversarial defenses

(both adversarial training and some certified defenses) and the distance between training data and test points. Additionally, we show that a tiny shift in input distribution (which may or may not be detectable in embedding space) can easily destroy the robustness property of an robust model.

## 3 METHODOLOGY

### 3.1 MEASURING THE DISTANCE BETWEEN TRAINING DATASET AND A TEST DATA POINT

To verify the correlation between the effectiveness of adversarial training and how close a test point is to the manifold of training dataset, we need to propose a reasonable distance metric between a test example and a set of training examples. However, defining a meaningful distance metric for high dimensional image data is a challenging problem. Naively using an Euclidean distance metric in the input space of images works poorly as it does not reflect the true distance between the images on their ground-truth manifold. One strategy is to use (kernel-)PCA, t-SNE (Maaten & Hinton, 2008), or UMAP (McInnes & Healy, 2018) to reduce the dimension of training data to a low dimensional space, and then define distance in that space. These methods are sufficient for small and simple datasets like MNIST, but for more general and complicated dataset like CIFAR, extracting a meaningful low-dimensional manifold directly on the input space can be really challenging.

On the other hand, using a DNN to extract features of input images and measuring the distance in the deep feature embedding space has demonstrated better performance in many applications (Hu et al., 2014; 2015), since DNN models can capture the manifold of image data much better than simple methods such as PCA or t-SNE. Although we can form an empirical distribution using kernel density estimation (KDE) on the deep feature embedding space and then obtain probability densities for test points, our experience showed that KDE work poorly in this case because the features extracted by DNNs are still high dimensional (hundreds or thousands dimensions).

Taking the above considerations into account, we propose a simple and intuitive distance metric using deep feature embeddings and $k$-nearest neighbour. Given a feature extraction neural network $h(x)$, a set of $n$ training data points $\mathcal{X}_{\text{train}} = \{x_{\text{train}}^1, x_{\text{train}}^2, \cdots, x_{\text{train}}^n\}$, and a set of $m$ test data points $\mathcal{X}_{\text{test}} = \{x_{\text{test}}^1, x_{\text{test}}^2, \cdots, x_{\text{test}}^m\}$ from the true data distribution, for each $j \in [m]$, we define the following distance between $x_{\text{test}}^j$ and $\mathcal{X}_{\text{train}}$:

$$D(x_{\text{test}}^j, \mathcal{X}_{\text{train}}) := \frac{1}{k} \sum_{i=1}^{k} \|h(x_{\text{test}}^j) - h(x_{\text{train}}^{\pi_j(i)})\|_p \tag{2}$$

where $\pi_j : [n] \to [n]$ is a permutation that $\{\pi_j(1), \pi_j(2), \cdots, \pi_j(n)\}$ is an ascending ordering of training data based on the $\ell_p$ distance between $x_{\text{test}}^j$ and $x_{\text{train}}^i$ in the deep embedding space, i.e.,

$$\forall i < i', \|h(x_{\text{test}}^j) - h(x_{\text{train}}^{\pi_j(i)})\|_p \leq \|h(x_{\text{test}}^j) - h(x_{\text{train}}^{\pi_j(i')})\|_p$$

In other words, we average the embedding space distance of $k$ nearest neighbors of $x_j$ in the training dataset. This simple metric is non-parametric and we found that the results are not sensitive to the selection of $k$; also, for naturally trained and adversarially trained feature extractors, the distance metrics obtained by different feature extractors reveal very similar correlations with the effectiveness of adversarial training.

### 3.2 MEASURING THE DISTANCE BETWEEN TRAINING AND TEST DATASETS

We are also interested to investigate the "distance" between the training dataset and the test dataset to gain some insights on how adversarial training performs on the entire test set. Unlike the setting in Section 3.1, this requires to compute a divergence between two empirical data distributions.

Given $n$ training data points $\mathcal{X}_{\text{train}} = \{x_{\text{train}}^1, x_{\text{train}}^2, \cdots, x_{\text{train}}^n\}$ and $m$ test data points $\mathcal{X}_{\text{test}} = \{x_{\text{test}}^1, x_{\text{test}}^2, \cdots, x_{\text{test}}^m\}$, we first apply a neural feature extractor $h$ to them, which is the same as in Section 3.1. Then, we apply a non-linear projection (in our case, we use t-SNE) to project both $h(x_{\text{train}}^j)$ and $h(x_{\text{test}}^j)$ to a low dimensional space, and obtain $\bar{x}_{\text{train}}^i = \text{proj}(h(x_{\text{train}}^i))$ and $\bar{x}_{\text{test}}^j = \text{proj}(h(x_{\text{test}}^j))$. The dataset after feature extraction and projection is denoted as $\bar{\mathcal{X}}_{\text{train}}$ and $\bar{\mathcal{X}}_{\text{test}}$. Because $\bar{x}_{\text{train}}^i$ and $\bar{x}_{\text{test}}^j$ are low dimensional, we can use kernel density estimation (KDE) to form empirical distributions $\bar{p}_{\text{train}}$ and $\bar{p}_{\text{test}}$ for them. We use $p_{\text{train}}$ and $p_{\text{test}}$ to denote the true distributions. Then, we

approximate the K-L divergence between $p_{\text{train}}$ and $p_{\text{test}}$ via a numerical integration of Eq.(3):

$$D_{\text{KL}}(p_{\text{train}}||p_{\text{test}}) \approx \int_V \bar{p}_{\text{train}}(x) \log \frac{\bar{p}_{\text{train}}(x)}{\bar{p}_{\text{test}}(x)} \mathrm{d}x, \tag{3}$$

where $\bar{p}_{\text{train}}(x) = \frac{1}{n}\sum_{i=1}^n K(x - \bar{x}_{\text{train}}^i; H)$ and $\bar{p}_{\text{test}}(x) = \frac{1}{m}\sum_{j=1}^m K(x - \bar{x}_{\text{test}}^j; H)$ are the KDE density functions. $K$ is the kernel function (specifically, we use the Gaussian kernel) and $H$ is the bandwidth parameter automatically selected by Scott's rule (Scott, 2015). $V$ is chosen as a box bounding all training and test data points. For a multi-class dataset, we compute the aforementioned KDE and K-L divergence *for each class separately*.

### 3.3 THE BLIND-SPOT ATTACK: A NEW CLASS OF ADVERSARIAL ATTACKS

Inspired by our findings of the negative correlation between the effectiveness of adversarial training and the distance between a test image and training dataset, we identify a new class of adversarial attacks called "blind-spot attacks", where we find input images that are "far enough" from any existing training examples such that:

- They are still drawn from the ground-truth data distribution (i.e. well recognized by humans) and *classified correctly* by the model (within the generalization capability of the model);
- Adversarial training cannot provide good robustness properties on these images, and we can easily find their adversarial examples with small distortions using a simple gradient based attack.

Importantly, blind-spot images are *not* adversarial images themselves. However, after performing adversarial attacks, we can find their adversarial examples with small distortions, despite adversarial training. In other words, we exploit the weakness in a model's robust generalization capability.

We find that these blind-spots are prevalent and can be easily found without resorting to complex generative models like in Song et al. (2018). For the MNIST dataset which Madry et al. (2018), Kolter & Wong (2018) and Sinha et al. (2018) demonstrate the strongest defense results so far, we propose a simple transformation to find the blind-spots in these models. We simply scale and shift each pixel value. Suppose the input image $x \in [-0.5, 0.5]^d$, we scale and shift each test data example $x$ element-wise to form a new example $x'$:

$$x' = \alpha x + \beta, \text{ s.t. } x' \in [-0.5, 0.5]^d$$

where $\alpha$ is a constant close to 1 and $\beta$ is a constant close to 0. We make sure that the selection of $\alpha$ and $\beta$ will result in a $x'$ that is still in the valid input range $[-0.5, 0.5]^d$. This transformation effectively adjusts the contrast of the image, and/or adds a gray background to the image. We then perform Carlini & Wagner's attacks on these transformed images $x'$ to find their adversarial examples $x'_{\text{adv}}$. It is important that the blind-spot images $x'$ are still undoubtedly valid images; for example, a digit that is slightly darker than the one in test set is still considered as a valid digit and can be well recognized by humans. Also, we found that with appropriate $\alpha$ and $\beta$ the accuracy of MNIST and Fashion-MNIST models barely decreases; the model has enough generalization capability for this set of slightly transformed images, yet their adversarial examples can be easily found.

Although the blind-spot attack is beyond the threat model considered in adversarial training (e.g. $\ell_\infty$ norm constrained perturbations), our argument is that adversarial training (and some other defense methods with certifications only on training examples such as Kolter & Wong (2018)) are unlikely to scale well to datasets that lie in a high dimensional manifold, as the limited training data only guarantees robustness near these training examples. The blind-spots are almost inevitable in high dimensional case. For example, in CIFAR-10, about 50% of test images are already in blind-spots and their adversarial examples with small distortions can be trivially found despite adversarial training. Using data augmentation may eliminate some blind-spots, however for high dimensional data it is impossible to enumerate all possible inputs due to the curse of dimensionality.

## 4 EXPERIMENTS

In this section we present our experimental results on adversarially trained models by Madry et al. (2018). Results on certified defense models by Kolter & Wong (2018); Wong et al. (2018) and Sinha et al. (2018) are very similar and are demonstrated in Section 6.4 in the Appendix.

## 4.1 SETUP

We conduct experiments on adversarially trained models by Madry et al. (2018) on four datasets: MNIST, Fashion MNIST, and CIFAR-10. For MNIST, we use the "secret" model release for the MNIST attack challenge[3]. For CIFAR-10, we use the public "adversarially trained" model[4]. For Fashion MNIST, we train our own model with the same model structure and parameters as the robust MNIST model, except that the iterative adversary is allowed to perturb each pixel by at most $\epsilon = 0.1$ as a larger $\epsilon$ will significantly reduce model accuracy.

We use our presented simple blind-spot attack in Section 3.3 to find blind-spot images, and use Carlini & Wagner's (C&W's) $\ell_\infty$ attack (Carlini & Wagner (2017)) to find their adversarial examples. We found that C&W's attacks generally find adversarial examples with smaller perturbations than projected gradient descent (PGD). To avoid gradient masking, we initial our attacks using two schemes: (1) from the original image plus a random Gaussian noise with a standard deviation of 0.2; (2) from a blank gray image where all pixels are initialized as 0. A successful attack is defined as finding an perturbed example that changes the model's classification and the $\ell_\infty$ distortion is less than a given $\epsilon$ used for robust training. For MNIST, $\epsilon = 0.3$; for Fashion-MNIST, $\epsilon = 0.1$; and for CIFAR, $\epsilon = 8/255$. All input images are normalized to $[-0.5, 0.5]$.

## 4.2 EFFECTIVENESS OF ADVERSARIAL TRAINING AND THE DISTANCE TO TRAINING SET

In this set of experiments, we build a connection between attack success rate on adversarially trained models and the distance between a test example and the whole training set. We use the metric defined in Section 3.1 to measure this distance. For MNIST and Fashion-MNIST, the outputs of the first fully connected layer (after all convolutional layers) are used as the neural feature extractor $h(x)$; for CIFAR, we use the outputs of the last average pooling layer. We consider both naturally and adversarially trained networks as the neural feature extractor, with $p = 2$ and $k = 5$. The results are shown in Figure 1, 2 and 3. For each test set, after obtaining the distance of each test point, we bin the test data points based on their distances to the training set and show them in the histogram at the bottom half of each figure (red). The top half of each figure (blue) represents the attack success rates for the test images in the corresponding bins. Some bars on the right are missing because there are too few points in the corresponding bins. We only attack correctly classified images and only calculate success rate on those images. Note that we should not compare the distances shown between the left and right columns of Figures 1, 2 and 3 because they are obtained using different embeddings, however the overall trends are very similar.

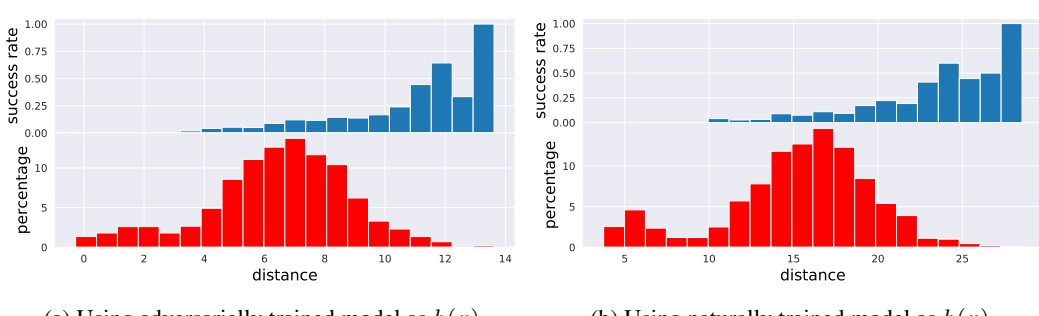

(a) Using adversarially trained model as $h(x)$      (b) Using naturally trained model as $h(x)$

Figure 1: Attack success rate and distance distribution of MNIST model in Madry et al. (2018). Upper: C&W $\ell_\infty$ attack success rates, $\epsilon = 0.3$. Lower: The distribution of the average $\ell_2$ (embedding space) distance between the images in test set and the top-5 nearest images in training set.

As we can observe in all three figures, most successful attacks in test sets for adversarially trained networks concentrate on the right hand side of the distance distribution, and the success rates tend to grow when the distance is increasing. The trend is independent of the feature extractor being used (naturally or adversarially trained). The strong correlation between attack success rates and the distance from a test point to the training dataset supports our hypothesis that adversarial training tends to fail on test points that are far enough from the training data distribution.

---

[3] https://github.com/MadryLab/mnist_challenge    [4] https://github.com/MadryLab/cifar10_challenge

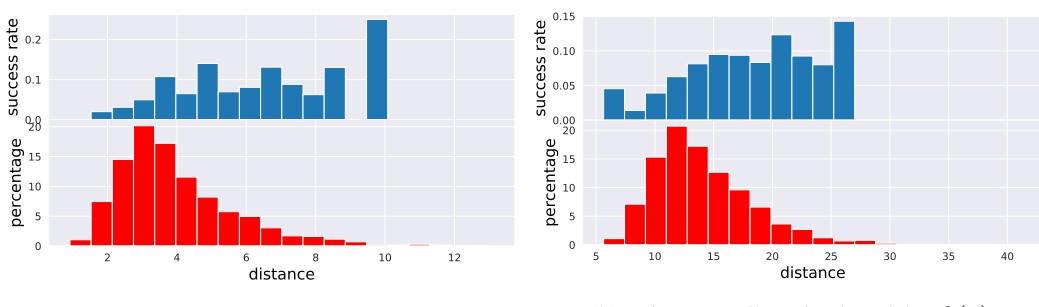

(a) Using adversarially trained model as $h(x)$      (b) Using naturally trained model as $h(x)$

Figure 2: Attack success rate and distance distribution of Fashion MNIST model trained using Madry et al. (2018). Upper: C&W $\ell_\infty$ attack success rate, $\epsilon = 0.1$. Lower: The distribution of the average $\ell_2$ (embedding space) distance between the images in test set and the top-5 nearest images in training set.

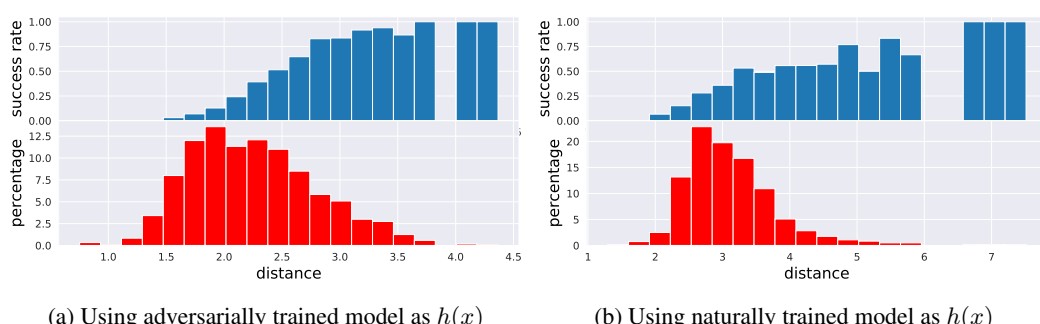

(a) Using adversarially trained model as $h(x)$      (b) Using naturally trained model as $h(x)$

Figure 3: Attack success rate and distance distribution of CIFAR model in Madry et al. (2018). Upper: C&W $\ell_\infty$ attack success rate, $\epsilon = 8/255$. Lower: The distribution of the average $\ell_2$ (embedding space) distance between the images in test set and the top-5 nearest images in training set.

### 4.3 K-L Divergence between training and test sets vs attack success rate

To quantify the overall distance between the training and the test set, we calculate the K-L divergence between the KDE distributions of training set and test set for each class according to Eq. (3). Then, for each dataset we take the average K-L divergence across all classes, as shown in Table 1. We use both adversarially trained networks and naturally trained networks as our feature extractors $h(x)$. Additionally, we also calculate the average normalized distance by calculating the $\ell_2$ distance between each test point and the training set as in Section 4.2, and taking the average over all test points. To compare between different datasets, we normalize each element of the feature representation $h(x)$ to mean 0 and variance 1. We average this distance among all test points and divide it by $\sqrt{d_t}$ to normalize the dimension, where $d_t$ is the dimension of the feature representation $h(\cdot)$.

Clearly, Fashion-MNIST is the dataset with the strongest defense as measured by the attack success rates on test set, and its K-L divergence is also the smallest. For CIFAR, the divergence between training and test sets is significantly larger, and adversarial training only has limited success. The hardness of training a robust model for MNIST is in between Fashion-MNIST and CIFAR. Another important observation is that the effectiveness of adversarial training *does not depend on the accuracy*; for Fashion-MNIST, classification is harder as the data is more complicated than MNIST, but training a robust Fashion-MNIST model is easier as the data distribution is more concentrated and adversarial training has less "blind-spots".

### 4.4 Blind-Spot Attack on MNIST and Fashion MNIST

In this section we focus on applying the proposed blind-spot attack to MNIST and Fashion MNIST. As mentioned in Section 3.3, for an image $x$ from the test set, the blind-spot image $x' = \alpha x + \beta$

Table 1: Average K-L divergence and normalized $\ell_2$ distance between training and test sets across all classes. We use both adversarially trained networks (adv.) and naturally trained networks (nat.) as our feature extractors when computing K-L divergence. Note that we only attack images that are correctly classified and report success rate on those images.

| Dataset | Avg K-L div. (adv. trained) | Avg K-L div. (nat. trained) | Avg. normalized $\ell_2$ Distance | Attack Success Rates (Test Set) | Test Accuracy |
|---|---|---|---|---|---|
| Fashion-MNIST | 0.046 | 0.058 | 0.4233 | 6.4% | 86.1% |
| MNIST | 0.119 | 0.095 | 0.3993 | 9.7% | 98.2% |
| CIFAR | 0.571 | 0.143 | 0.6715 | 37.9% | 87.0% |

obtained by scaling and shifting is considered as a new natural image, and we use the C&W $\ell_\infty$ attack to craft an adversarial image $x'_{\text{adv}}$ for $x'$. The attack distortion is calculated as the $\ell_\infty$ distance between $x'$ and $x'_{\text{adv}}$. For MNIST, $\epsilon = 0.3$ so we set the scaling factor to $\alpha = \{1.0, 0.9, 0.8, 0.7\}$. For Fashion-MNIST, $\epsilon = 0.1$ so we set the scaling factor to $\alpha = \{1.0, 0.95, 0.9\}$. We set $\beta$ to either 0 or a small constant. The case $\alpha = 1.0, \beta = 0.0$ represents the original test set images. We report the model's accuracy and attack success rates for each choice of $\alpha$ and $\beta$ in Table 2 and Table 3. Because we scale the image by a factor of $\alpha$, we also set a stricter criterion of success – the $\ell_\infty$ perturbation must be less than $\alpha\epsilon$ to be counted as a successful attack. For MNIST, $\epsilon = 0.3$ and for Fashion-MNIST, $\epsilon = 0.1$. We report both success criterion, $\epsilon$ and $\alpha\epsilon$ in Tables 2 and 3.

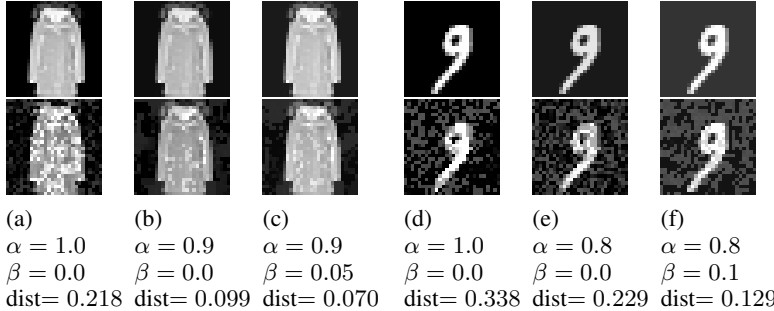

(a)
$\alpha = 1.0$
$\beta = 0.0$
dist= 0.218

(b)
$\alpha = 0.9$
$\beta = 0.0$
dist= 0.099

(c)
$\alpha = 0.9$
$\beta = 0.05$
dist= 0.070

(d)
$\alpha = 1.0$
$\beta = 0.0$
dist= 0.338

(e)
$\alpha = 0.8$
$\beta = 0.0$
dist= 0.229

(f)
$\alpha = 0.8$
$\beta = 0.1$
dist= 0.129

Figure 4: Blind-spot attacks on Fashion-MNIST and MNIST data with scaling and shifting on adversarially trained models (Madry et al., 2018). First row contains input images after scaling and shifting and the second row contains the found adversarial examples. "dist" represents the $\ell_\infty$ distortion of adversarial perturbations. The first rows of figures (a) and (d) represent the original test set images ($\alpha = 1.0, \beta = 0.0$); first rows of figures (b), (c), (e), and (f) illustrate the images after transformation. Adversarial examples for these transformed images have small distortions.

We first observe that for all pairs of $\alpha$ and $\beta$ the transformation does not affect the models' test accuracy at all. The adversarially trained model classifies these slightly scaled and shifted images very well, with test accuracy equivalent to the original test set. Visual comparisons in Figure 4 show that when $\alpha$ is close to 1 and $\beta$ is close to 0, it is hard to distinguish the transformed images from the original images. On the other hand, according to Tables 2 and 3, the attack success rates for those transformed test images are significantly higher than the original test images, for both the original criterion $\epsilon$ and the stricter criterion $\alpha\epsilon$. In Figure 4, we can see that the $\ell_\infty$ adversarial perturbation

| $\alpha, \beta$ | $\alpha = 1.0$ $\beta = 0$ | $\alpha = 0.9$ $\beta = 0$ | $\beta = 0.05$ | $\alpha = 0.8$ $\beta = 0$ | $\beta = 0.1$ | $\alpha = 0.7$ $\beta = 0$ | $\beta = 0.15$ |
|---|---|---|---|---|---|---|---|
| acc | 98.2% | 98.3% | 98.5% | 98.4% | 98.5% | 98.4% | 98.1% |
| th. | 0.3 | 0.3 | 0.27 | 0.3 | 0.27 | 0.3 | 0.24 | 0.3 | 0.24 | 0.3 | 0.21 | 0.3 | 0.21 |
| suc. rate | 9.70% | 75.20% | 15.20% | 93.65% | 82.50% | 94.85% | 52.30% | 99.55% | 95.45% | 98.60% | 82.45% | 99.95% | 99.95% |

Table 2: Attack success rate (suc. rate) and test accuracy (acc) of scaled and shifted MNIST. An attack is considered successful if its $\ell_\infty$ distortion is less than thresholds (th.) 0.3 or $0.3\alpha$.

| $\alpha, \beta$ | $\alpha = 1.0$ | $\alpha = 0.95$ | | | | $\alpha = 0.9$ | | | |
| --- | --- | --- | --- | --- | --- | --- | --- | --- | --- |
| | $\beta = 0$ | $\beta = 0$ | | $\beta = 0.025$ | | $\beta = 0$ | | $\beta = 0.05$ | |
| acc | 86.1% | 86.1% | | 86.4% | | 86.1% | | 86.2% | |
| th. | 0.1 | 0.1 | 0.095 | 0.1 | 0.095 | 0.1 | 0.09 | 0.1 | 0.09 |
| suc. rate | 6.40% | 11.25% | 9.05% | 22.55% | 18.55% | 25.70% | 19.15% | 62.60% | 55.95% |

Table 3: Attack success rate (suc. rate) and test accuracy (acc) of scaled and shifted Fashion-MNIST. An attack is considered as successful if its $\ell_\infty$ distortion is less than threshold (th.) 0.1 or $0.1\alpha$.

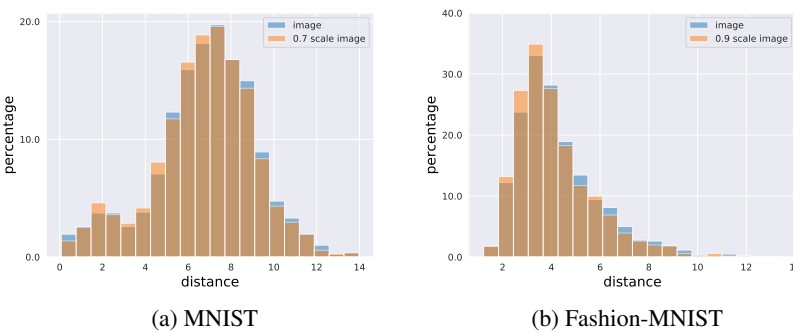

(a) MNIST         (b) Fashion-MNIST

Figure 5: The distribution of the $\ell_2$ distance between the original and scaled images in test set and the top-5 nearest images ($k = 5$) in training set using the distance metric defined in Eq. (2).

required is much smaller than the original image after the transformation. Thus, the proposed scale and shift transformations indeed move test images into blind-spots. More figures are in Appendix.

One might think that we can generally detect blind-spot attacks by observing their distances to the training dataset, using a metric similar to Eq. (2). Thus, we plot histograms for the distances between tests points and training dataset, for both original test images and those slightly transformed ones in Figure 5. We set $\alpha = 0.7, \beta = 0$ for MNIST and $\alpha = 0.9, \beta = 0$ for Fashion-MNIST. Unfortunately, the differences in distance histograms for these blind-spot images are so tiny that we cannot reliably detect the change, yet the robustness property drastically changes on these transformed images.

## 5 CONCLUSION

In this paper, we observe that the effectiveness of adversarial training is highly correlated with the characteristics of the dataset, and data points that are far enough from the distribution of training data are prone to adversarial attacks despite adversarial training. Following this observation, we defined a new class of attacks called "blind-spot attack" and proposed a simple scale-and-shift scheme for conducting blind-spot attacks on adversarially trained MNIST and Fashion MNIST datasets with high success rates. Our findings suggest that adversarial training can be challenging due to the prevalence of blind-spots in high dimensional datasets.

## ACKNOWLEDGMENT

We thank Zeyuan Allen-Zhu, Lijie Chen, Sébastien Bubeck, Rasmus Kyng, Yin Tat Lee, Aleksander Mądry, Jelani Nelson, Eric Price, Ilya Razenshteyn, Aviad Rubinstein, Ludwig Schmidt and Pengchuan Zhang for fruitful discussions. We also thank Eric Wong for kindly providing us with pre-trained models to perform our experiments.

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

## 6 APPENDIX

### 6.1 DISTANCE DISTRIBUTIONS UNDER DIFFERENT NEAREST NEIGHBOUR PARAMETERS $k$

As discussed in Section 3.1, we use $k$-nearest neighbour in embedding space to measure the distance between a test example and the training set. In Section 4.2 we use $k = 5$. In this section we show that the choice of $k$ does not have much influence on our results. We use the adversarially trained model on the CIFAR dataset as an example. In Figures 6, 7 and 8 we choose $k = 10, 100, 1000$, respectively. The results are similar to those we have shown in Figure 3: a strong correlation between attack success rates and the distance from a test point to the training dataset.

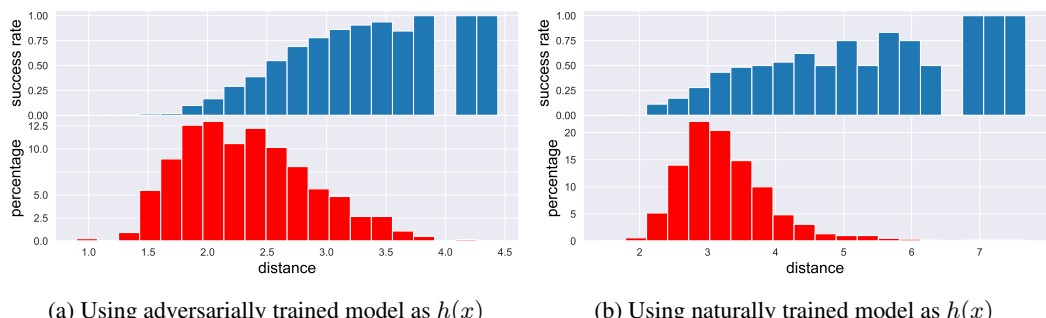

(a) Using adversarially trained model as $h(x)$     (b) Using naturally trained model as $h(x)$

Figure 6: Attack success rates and distance distribution of the adversarially trained CIFAR model by Madry et al. (2018). Upper: C&W $\ell_\infty$ attack success rate, $\epsilon = 8/255$. Lower: distribution of the average $\ell_2$ (embedding space) distance between the images in test set and the **top-10** ($k = 10$) nearest images in training set.

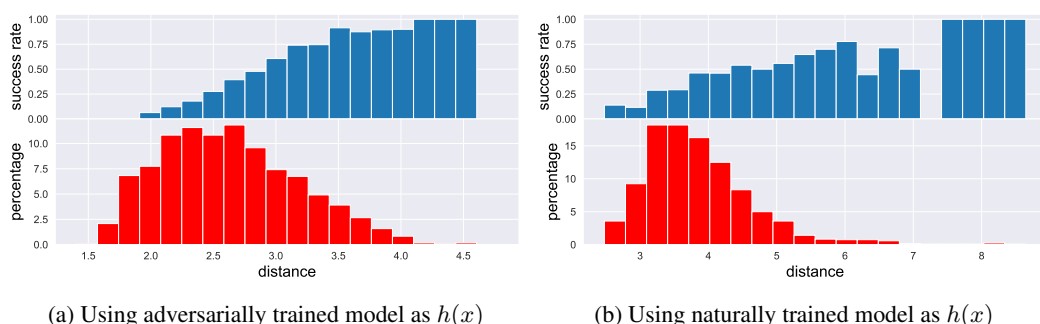

(a) Using adversarially trained model as $h(x)$     (b) Using naturally trained model as $h(x)$

Figure 7: Attack success rates and distance distribution of the adversarially trained CIFAR model by Madry et al. (2018). Upper: C&W $\ell_\infty$ attack success rate, $\epsilon = 8/255$. Lower: distribution of the average $\ell_2$ (embedding space) distance between the images in test set and the **top-100** ($k = 100$) nearest images in training set.

### 6.2 GERMAN TRAFFIC SIGN (GTS) DATASET

We also studied the German Traffic Sign (GTS) (Houben et al., 2013) dataset. For GTS, we train our own model with the same model structure and parameters as the adversarially trained CIFAR model (Madry et al., 2018). We set $\epsilon = 8/255$ for adversarial training with PGD, and also use the same $\epsilon$ as the threshold of success. The results are shown in Figure 9. The GTS model behaves similarly to the CIFAR model: attack success rates are much higher when the distances between the test example and the training dataset are larger.

### 6.3 MORE VISUALIZATION RESULTS

We demonstrate more MNIST and Fashion-MNIST visualizations in Figure 10.

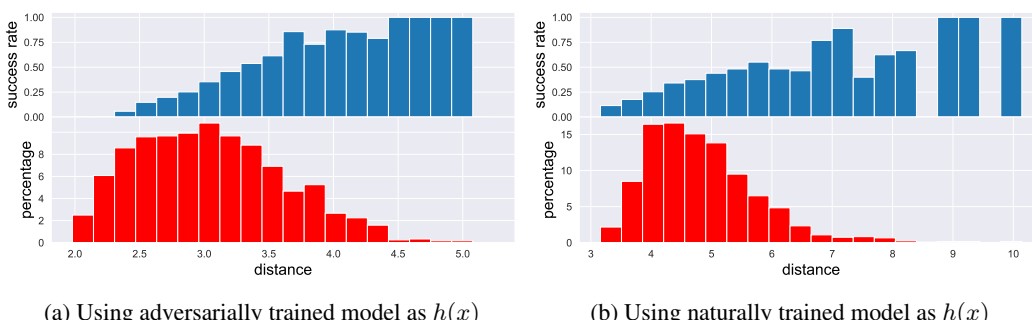

(a) Using adversarially trained model as $h(x)$      (b) Using naturally trained model as $h(x)$

Figure 8: Attack success rates and distance distribution of the adversarially trained CIFAR model by Madry et al. (2018). Upper: C&W $\ell_\infty$ attack success rate, $\epsilon = 8/255$. Lower: distribution of the average $\ell_2$ (embedding space) distance between the images in test set and the **top-1000** ($k = 1000$) nearest images in training set.

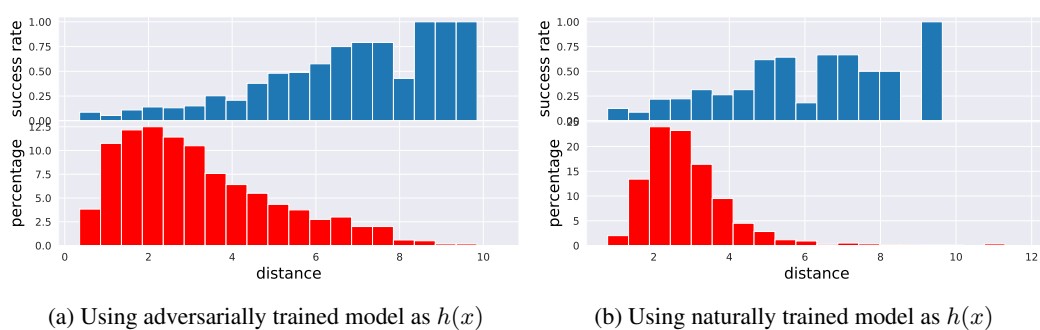

(a) Using adversarially trained model as $h(x)$      (b) Using naturally trained model as $h(x)$

Figure 9: Attack success rate and distance distribution of GTS in Madry et al. (2018). Upper: C&W $\ell_\infty$ attack success rate, $\epsilon = 8/255$. Lower: distribution of the average $\ell_2$ (embedding space) distance between the images in test set and the top-5 nearest images in training set.

## 6.4 RESULTS ON OTHER ROBUST TRAINING METHODS

In this section we demonstrate our experimental results on two other state-of-the-art *certified* defense methods, including convex adversarial polytope by Wong et al. (2018) and Kolter & Wong (2018), and distributional robust optimization based adversarial training by Sinha et al. (2018). Different from the adversarial training by Madry et al. (2018), these two methods can provide a formal certification on the robustness of the model and provably improve robustness on the training dataset. However, they cannot practically guarantee non-trivial robustness on test data. We did not include other certified defenses like Raghunathan et al. (2018) and Hein & Andriushchenko (2017) because they are not applicable to multi-layer networks. For all defenses, we use their official implementations and pretrained models (if available). Figure 11 shows the results on CIFAR using the defenses in Wong et al. (2018). Tables 4 and 5 show the blind-spot attack results on MNIST and Fashion-MNIST for robust models in Kolter & Wong (2018) and Sinha et al. (2018), respectively. Figure 12 shows the blind-spot attack examples on Madry et al. (2018), Kolter & Wong (2018) and Sinha et al. (2018).

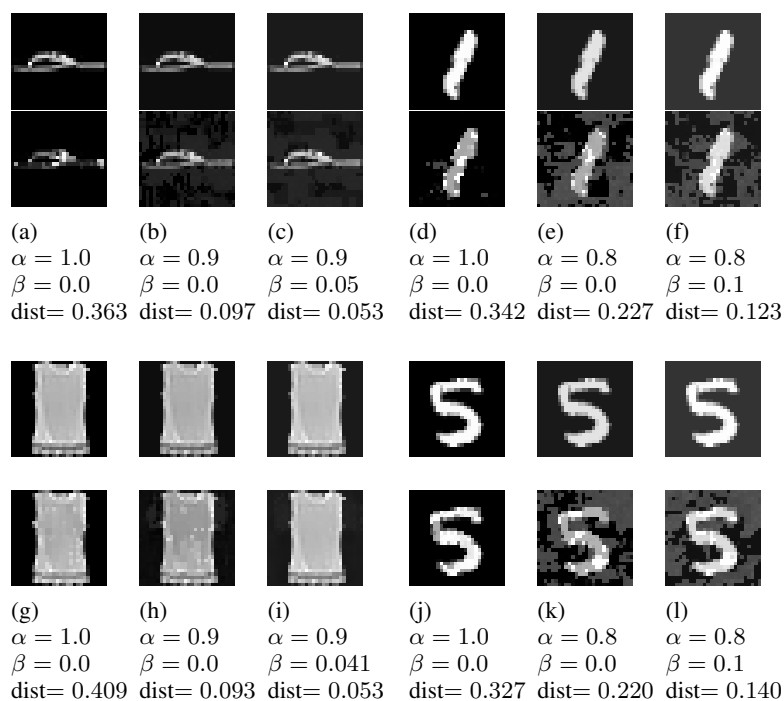

Figure 10: Blind-spot attacks on Fashion-MNIST and MNIST data with scaling and shifting in Madry et al. (2018). First row contains input images after scaling and shifting and the second row contains the found adversarial examples. "dist" represents the $\ell_\infty$ distortion of adversarial perturbations. The first rows of figures (a), (d), (g) and (j) represent the original test set images ($\alpha = 1.0, \beta = 0.0$); first rows of figures (b), (c), (e), (f), (h), (i), (k) and (l) illustrate the images after transformation. Adversarial examples for these transformed images can be found with small distortions.

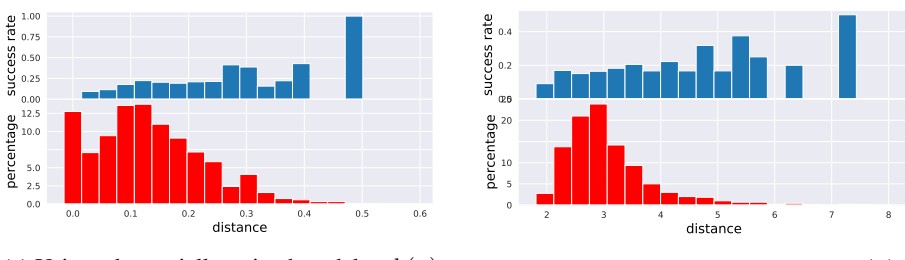

(a) Using adversarially trained model as $h(x)$     (b) Using naturally trained model as $h(x)$

Figure 11: Attack success rates and distance distribution of the CIFAR-10 model in Wong et al. (2018). Lower: the histogram of the average $\ell_2$ (in embedding space) distance between the images in test set and the top-5 nearest images in training set. Upper: the C&W $\ell_\infty$ attack success rate with success criterion $\epsilon = 8/255$.

| | $\alpha, \beta$ | $\alpha = 1.0$ | $\alpha = 0.95$ | | | | $\alpha = 0.9$ | | | |
| --- | --- | --- | --- | --- | --- | --- | --- | --- | --- | --- |
| | | $\beta = 0$ | $\beta = 0$ | | $\beta = 0.025$ | | $\beta = 0$ | | $\beta = 0.05$ | |
| MNIST | Accuracy | 97.5% | 97.5% | | 97.5% | | 97.5% | | 97.4% | |
| | Success criterion ($\ell_\infty$ norm) | 0.1 | 0.1 | 0.095 | 0.1 | 0.095 | 0.1 | 0.09 | 0.1 | 0.09 |
| | Success rates | 2.15% | 5.55% | 4.35% | 28.5% | 17.55% | 30.1% | 15.4% | **86.35%** | 80.7% |
| | $\alpha, \beta$ | $\alpha = 1.0$ | $\alpha = 0.95$ | | | | $\alpha = 0.9$ | | | |
| | | $\beta = 0$ | $\beta = 0$ | | $\beta = 0.025$ | | $\beta = 0$ | | $\beta = 0.05$ | |
| Fashion-MNIST | Accuracy | 79.1% | 79.1% | | 79.4% | | 79.2% | | 79.2% | |
| | Success criterion ($\ell_\infty$ norm) | 0.1 | 0.1 | 0.095 | 0.1 | 0.095 | 0.1 | 0.09 | 0.1 | 0.09 |
| | Success rates | 6.85% | 15.45% | 9.3% | 39.75% | 29.35% | 34.25% | 24.65% | **69.95%** | 65.2% |

Table 4: Blind-spot attack on MNIST and Fashion-MNIST for robust models by Kolter & Wong (2018)

| MNIST | $\alpha, \beta$ | $\alpha=1.0$ | $\alpha=0.95$ | | | | $\alpha=0.9$ | | | |
|---|---|---|---|---|---|---|---|---|---|---|
| | | $\beta=0$ | $\beta=0$ | | $\beta=0.025$ | | $\beta=0$ | | $\beta=0.05$ | |
| | Accuracy | 98.7% | 98.5% | | 98.6% | | 98.7% | | 98.4% | |
| | Success criterion ($\ell_2$ norm) | 2 | 2 | 1.9 | 2 | 1.9 | 2 | 1.8 | 2 | 1.8 |
| | Success rates | 12.2% | 27.05% | 22.95% | 36.15% | 30.9% | 45.25% | 31.55% | **58.9%** | 45.6% |
| Fashion-MNIST | $\alpha, \beta$ | $\alpha=1.0$ | $\alpha=0.95$ | | | | $\alpha=0.9$ | | | |
| | | $\beta=0$ | $\beta=0$ | | $\beta=0.025$ | | $\beta=0$ | | $\beta=0.05$ | |
| | Accuracy | 88.5% | 88.3% | | 88.2% | | 88.1% | | 87.8% | |
| | Success criterion ($\ell_2$ norm) | 2 | 2 | 1.9 | 2 | 1.9 | 2 | 1.8 | 2 | 1.8 |
| | Success rates | 31.4% | 46.3% | 41.1% | 58 % | 53.3% | 61.2% | 51.8% | **69.1%** | 62.85% |

Table 5: Blind-spot attack on MNIST and Fashion-MNIST for robust models by Sinha et al. (2018). Note that we use $\ell_2$ distortion for this model as it is the threat model under study in their work.

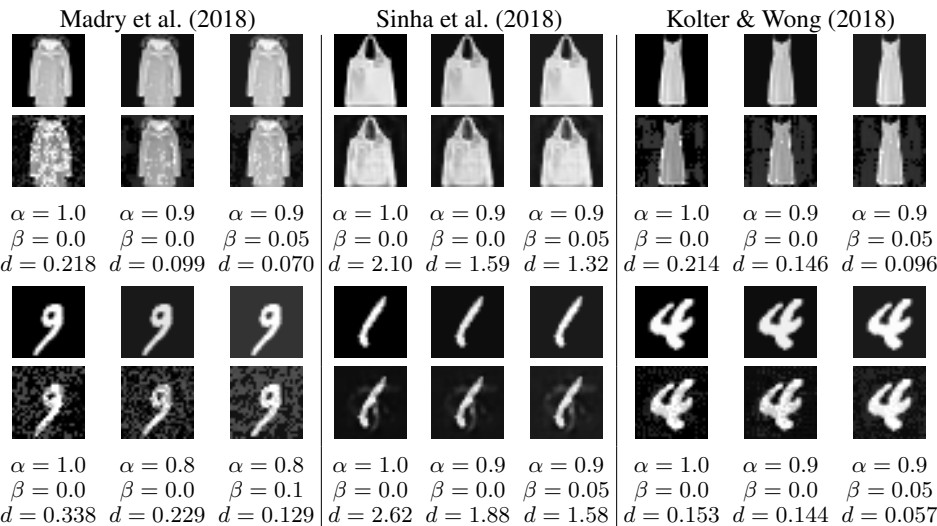

Figure 12: Blind-spot attacks on Fashion-MNIST and MNIST datasets with scaling and shifting. For each group, the first row contains input images transformed with different scaling and shifting parameter $\alpha, \beta$ ($\alpha=1.0, \beta=0.0$ is the original image) and the second row contains the found adversarial examples. $d$ represents the distortion of adversarial perturbations. For models from Madry et al. (2018) and Kolter & Wong (2018) we use $\ell_\infty$ norm and for models from Sinha et al. (2018) we use $\ell_2$ norm. Adversarial examples for these transformed images can be found with small distortions $d$.

