# OpenReview forum: "The Limitations of Adversarial Training and the Blind-Spot Attack"
_ICLR.cc/2019/Conference_

### Official Review · AnonReviewer3 · 2018-10-30
**Reviewer's summery: interesting idea/findings but with questions**

**Rating:** 6
**Confidence:** 4

**Review:**

In this paper, the authors associated with the generalization gap of robust adversarial training with the distance between the test point and the manifold of training data. A so-called 'blind-spot attack' is proposed to show the weakness of robust adversarial training.  Although the paper contains interesting ideas and empirical results, I have several concerns about the current version.

a) In the paper, the authors mentioned that "This simple metric is non-parametric and we found that the results are not sensitive to the selection of k". Can authors provide more details, e.g., empirical results, about it? What is its rationale?

b) In the paper, "We find that these blind-spots are prevalent and can be easily found without resorting to complex
generative models like in Song et al. (2018). For the MNIST dataset which Madry et al. (2018) demonstrate the strongest defense results so far, we propose a simple transformation to find the blind-spots in this model." Can authors provide empirical comparison between blind-spot attacks and the work by Song et al. (2018), e.g., attack success rate & distortion?

c) The linear transformation x^\prime = \alpha x + \beta yields a blind-spot attack which can defeat robust adversarial training. However, given the linear transformation, one can further modify the inner maximization (adv. example generation) in robust training framework so that the $\ell_infty$ attack satisfies  max_{\alpha, \beta} f(\alpha x + \beta) subject to \| \alpha x + \beta \|\leq \epsilon. In this case, robust training framework can defend blind-spot attacks, right? I agree with the authors that the generalization error is due to the mismatch between training data and test data distribution, however, I am not convinced that blind-spot attacks are effective enough to robust training.

d) "Because we scale the image by a factor of \alpha, we also set a stricter criterion of success, ..., perturbation must be less
than \alpha \epsilon to be counted as a successful attack." I did not get the point. Even if you have a scaling factor in x^\prime = \alpha x + \beta, the universal perturbation rule should still be | x - x^\prime  |_\infty \leq \epsilon. The metric the authors used would result in a higher attack success rate, right?

---

> ### Author Response · Authors · 2018-11-09
> **Thank you for the questions! We have updated our paper and answered your questions below.**
>
> Dear AnonReviewer3,
>
> Thank you for your insightful questions. They are very helpful for us to improve the paper. We would like to answer your 4 questions as below.
>
> a) We added more figures with k=10, 100, 1000 in the appendix (in main text, we used k=5). Our main conclusion does not change regardless the value of k: there is a strong correlation between attack success rate and the distance between test examples to training dataset. A larger distance usually implies a higher attack success rate. The rational to use this metric is that it is simple, and nearest neighbour based methods are usually robust to hyper-parameter selection. We don’t want our observations depend on hyper-parameters during distance measurement.
>
> b) Song et al. (2018) does not have ordinary metrics like distortion or (ordinary) attack success rates to compare with. In their attack, the input is a random noise for GAN, and they generate  adversarial images from scratch. In typical adversarial attacks, people start from a specific reference (natural) image x and add adversarial distortion to obtain x_adv. In their paper, adversarial images are generated by GANs directly and there is no reference images at all, so distortion cannot be calculated (see definitions 1 and 2 in their paper). They have to conduct user study to determine what is the true class label for a generated image, and see if the model will misclassify it. The success rate is the model’s misclassification rate from user study.
>
> In our paper, our attacks first conduct slight transformations on a natural test image x to obtain x’, and then run ordinary gradient based adversarial attacks on x’ to obtain x’_adv. We have a reference image x’, so we can compute the distortion between x’ and x’_adv, and determine the success by a certain criterion on distortion. This setting is different from Song et al. (2018) so we cannot directly compare distortion and success rates with them.
>
> c) We want to emphasize that the “blind-spot attack” is a class of attacks, which exploits the gap between training and test data distributions (see our definition in Section 3.3). The linear transformation used in our paper is one of the simplest attacks in this class. If we know the details of this specific attack before training, it is possible defend against this specific simple attack. However, it is always possible to find some different blind-spot attacks (for example, by using a generative model). Rather than starting a new arm race between attacks and defenses, our argument here is to show the fundamental limitations of adversarial training -- it is hard to cover all the blind-spots during training time because it is impossible to eliminate the gap between training and test data especially when data dimension is high.
>
> d) The stricter criterion actually makes our attack success rates *lower* rather than higher. Finding adversarial examples with smaller distortions is harder than finding adversarial examples with large distortions. As an extreme case, if the criterion is distortion<=0, the attack success rate will always be zero, since we cannot fool the model using unmodified natural images. In Table 2, the success rates under the column 0.27 are strictly lower than the numbers under the column 0.3. We consider this additional stricter criterion because images after scaling are within a smaller range, so we also restrict the noise to be smaller, to keep the same signal-to-noise ratio and make an absolutely fair comparison. If we don’t use this stricter criterion, our attack success rates will look even better.
>
>
> In our updated revision, we also include additional experiments on GTS dataset, as long as two other state-of-the-art adversarial training methods by Wong et al. and Sinha et al.. We observe very similar results on all these methods and datasets, further confirming the conclusion of our paper.
>
> We hope our answers resolve all the doubts you had with our paper. We would like to further discuss with you if you have any unclear things or additional questions, and hope you can reconsider the rating of our paper.
>
> Thank you!
> Paper 1584 Authors

---

> > ### Author Response · Authors · 2018-11-22
> > **We will really appreciate it if you could provide us more feedback before the revision period ends**
> >
> > Dear AnonReviewer3,
> >
> > Thank you again for your insightful and constructive comment!
> >
> > We hope that we have addressed your questions. We understand you may be discussing our paper with other reviewers and you can take your time. As the revision period is closing soon, we will really appreciate it if you could let us know if you find anything unclear in our response, or have any further concerns about our paper. We will try our best to revise our paper based on your suggestions before the revision period ends.
> >
> > Thank you!
> > Paper 1584 Authors

---

> > > ### Comment · AnonReviewer3 · 2018-11-26
> > > **Responses clarified the reviewer's previous questions**
> > >
> > > "We want to emphasize that the “blind-spot attack” is a class of attacks, which exploits the gap between training and test data distributions (see our definition in Section 3.3). The linear transformation used in our paper is one of the simplest attacks in this class. If we know the details of this specific attack before training, it is possible defend against this specific simple attack."
> > >
> > > Ok, I agree with the authors at this point.
> > >
> > > "The stricter criterion actually makes our attack success rates *lower* rather than higher. Finding adversarial examples with smaller distortions is harder than finding adversarial examples with large distortions. As an extreme case, if the criterion is distortion<=0, the attack success rate will always be zero, since we cannot fool the model using unmodified natural images. In Table 2, the success rates under the column 0.27 are strictly lower than the numbers under the column 0.3. We consider this additional stricter criterion because images after scaling are within a smaller range, so we also restrict the noise to be smaller, to keep the same signal-to-noise ratio and make an absolutely fair comparison. If we don’t use this stricter criterion, our attack success rates will look even better.
> > > "
> > >
> > > Yes, the authors are correct that finding adversarial examples with smaller distortions is harder than finding adversarial examples with large distortions, thus $\alpha \epsilon$ will make attack success rate (ASR) LOWER. Based on that, I checked Table 2, which is still unclear to me.
> > >
> > > In the last column of Table 2, alpha = 0.7 & beta = 0.15, I wonder why ASRs under thr = 0.3 and thr = 0.21 are the same. Since an attack is considered as successful if its Linf distortion is less than given thrs, I assumed that the distortion condition will be examined as $| \alpha x + \beta - x |_infty \leq \eps$, right? If so, it quite surprising that ASRs for the two cases (alpha = 0.7, beta = 0, thr = 0.21) and  (alpha = 0.7, beta = 0.15, thr = 0.21) have a large gap. Any rationale behind that?
> > >
> > >
> > > I will adjust my score based on the authors' further clarification.

---

> > > > ### Author Response · Authors · 2018-11-26
> > > > **Thank you for the questions! Here are our further clarifications.**
> > > >
> > > > Dear AnonReviewer3,
> > > >
> > > > Thank you for your response and further questions. We would like to answer them as below:
> > > >
> > > > “I assumed that the distortion condition will be examined as $| \alpha x + \beta |_infty \leq \eps$, right?”
> > > > No, this is not how we examine the Linf distortion success condition in Table 2.
> > > >
> > > > We use alpha and beta to obtain new natural reference images instead of adversarial images. For example, for an original image x from the test set, we scale and shift this image to obtain a new natural reference image x’ = \alpha * x + \beta. Then we run C&W attack on x’ to obtain its adversarial image x’_adv. Note that x’ = \alpha * x + \beta is not considered as an adversarial image but as a natural image since in the blind-spot attack we are finding the blind-spots (where the model do not have good robustness) in the natural data distribution.
> > > >
> > > > The distortion condition is examined as the distance between x’ and x’_adv: $|x’ - x’_adv|_\infty \leq \eps$, but not $| \alpha x + \beta |_infty \leq \eps$. We will try to make this clearer in our revision.
> > > >
> > > > “In the last column of Table 2, alpha = 0.7 & beta = 0.15, I wonder why ASRs under thr = 0.3 and thr = 0.21 are the same.”
> > > > The reason is that most adversarial examples generated from blind-spot images with alpha=0.7 and beta=0.15 have small distortions, less than both 0.3 and 0.21. So they are considered successful in both criteria.
> > > >
> > > > “it quite surprising that ASRs for the two cases (alpha = 0.7, beta = 0, thr = 0.21) and  (alpha = 0.7, beta = 0.15, thr = 0.21) have a large gap. Any rationale behind that?”
> > > > The ASR for the case with non-zero beta is much higher than beta=0 case indicates that scaling+shifting is more effective than scaling alone to reduce the robustness of the model under attack. Scaling+shifting is a more powerful blind-spot attack.
> > > >
> > > > We are glad to discuss further with you if you have any additional questions. Thanks again for the constructive feedback!
> > > >
> > > > Thank you!
> > > > Paper 1584 Authors

---

> > > > > ### Comment · AnonReviewer3 · 2018-11-26
> > > > > **Further discussion on 'blind-spot' attack**
> > > > >
> > > > > Thanks for the clarification.
> > > > >
> > > > > "We use alpha and beta to obtain new natural reference images instead of adversarial images."
> > > > >
> > > > > This is a key point which makes reviewer confusing, since in Sec. 3.3, blind-spot attacks seem to generate adversarial images only using scaling and shifting. However, in experiments x’ = \alpha * x + \beta is used to generate natural reference image. I am Okay with that only if the experiment is consistent, e.g., applying C\&W attack on x’ = \alpha * x + \beta for all \alpha and \beta discussed in this paper.
> > > > >
> > > > > Please carefully revise Sec. 3.3. and experiment section to make the aforementioned point clearer.
> > > > >
> > > > > Based on the authors's current response, I increase my score to 6.

---

> > > > > > ### Author Response · Authors · 2018-11-26
> > > > > > **Thank you so much for your comments and considerations**
> > > > > >
> > > > > > We really appreciate the reviewer's fruitful suggestions, and we see where the confusion is.
> > > > > >
> > > > > > In Sec. 3.3, blind-spot attack uses scaling and shifting to generate new natural reference images x' = \alpha * x + \beta. We still apply C\&W L_inf attacks on x’ to generate adversarial images x'_adv for all \alpha and \beta. We will revise our paper to make this clearer.
> > > > > >
> > > > > > Thank you again for your comments and we will make the writing better.
> > > > > >
> > > > > > Thank you!
> > > > > > Paper 1584 Authors

---

### Official Review · AnonReviewer1 · 2018-11-03
**Clear and simple idea, insightful experiments.**

**Rating:** 7
**Confidence:** 2

**Review:**

The paper is well written and the main contribution, a methodology to find “blind-spot attacks” well motivated and differences to prior work stated clearly.

The empirical results presented in Figure 1 and 2 are very convincing. The gain of using a sufficiently more complicated approach to assess the overall distance between the test and training dataset is not clear, comparing it to the very insightful histograms. Why for example not using a simple score based on the histogram, or even the mean distance? Of course providing a single measure would allow to leverage that information during training. However, in its current form this seems rather complicated and computationally expensive (KL-based). As stated later in the paper the histograms themselves are not informative enough to detect such blind-spot transformation. Intuitively this makes a lot of sense given that the distance is based on the network embedding and is therefore also susceptible to this kind of data. However, it is not further discussed how the overall KL-based data similarity measure would help in this case since it seems likely that it would also exhibit the same issue.

---

> ### Author Response · Authors · 2018-11-19
> **Thank you for the questions! We have updated our paper and answered your questions below.**
>
> Thank you for the encouraging comments. First of all, we would like to mention that we add more experiments on two additional state-of-the-art strong and certified defense methods, and observe that they are also vulnerable to blind-spot attacks. Please see our reply to all reviewers.
>
> We agree that the K-L based method is complicated and computationally extensive. Fortunately, we only need to compute it once per dataset. To the best of our knowledge, currently, there is no perfect metric to measure the distance between a training set and a test set. Ordinary statistical methods (like kernel two-sample tests) do not work well due to the high dimensionality and the complex nature of image data. So the measurement we proposed is a best-effort attempt that can hopefully give us some insights into this problem.
>
> As suggested by the reviewer, we added a new metric based on the mean of \ell_2 distance on the histogram in Section 4.3. The results are shown in Table 1 (under column “Avg. normalized l2 Distance”). The results align well with our conclusion: the dataset with significant better attack success rates has noticeably larger distance. It further supports the conclusion of our paper and indicates that our conclusion is distance metric agnostic.
>
> We hope that we have made everything clear, and we again appreciate your comments. Let us know if you have any additional questions.
>
> Thank you!
> Paper 1584 Authors

---

### Official Review · AnonReviewer2 · 2018-11-04
**An interesting paper analyzing the effect of the distance between training and test set on robustness of adversarial training**

**Rating:** 7
**Confidence:** 3

**Review:**

This paper provides some insights on influence of data distribution on robustness of adversarial training. The paper demonstrates through a number of analysis that the distance between the training an test data sets plays an important role on the effectiveness of adversarial training. To show the latter, the paper proposes an approach to measure the distance between the two data sets using combination of nonlinear projection (e.g. t-SNE), KDE, and K-L divergence. The paper also shows that under simple transformation to the test dataset (e.g. scaling), performance of adversarial training reduces significantly due to the large gap between training and test data set. This tends to impact high dimensional data sets more than low dimensional data sets since it is much harder to cover the whole ground truth data distribution in the training dataset.

Pros:
- Provides insights on why adversarial training is less effective on some datasets.
- Proposes a metric that seems to strongly correlate with the effectiveness of adversarial training.

Cons:
- Lack of theoretical analysis. It could have been nice if the authors could show the observed phenomenon analytically on some simple distribution.
- The marketing phrase "the blind-spot attach" falls short in delivering what one may expect from the paper after reading it. The paper would read much better if the authors better describe the phenomena based on the gap between the two distribution than using bling-spot. For some dataset, this is beyond a spot, it could actually be huge portion of the input space!

Minor comments:
- I believe one should not compare the distance shown between the left and right columns of Figure 3 as they are obtained from two different models. Though the paper is not suggesting that, it would help to clarify it in the paper. Furthermore, it would help if the paper elaborates why the distance between the test and training dataset is smaller in an adversarially trained network compared to a naturally trained network.
- Are the results in Table 1 for an adversarially trained network or a naturally trained network? Either way, it could be also interesting to see the average K-L divergence between an adversarially and a naturally trained network on the same dataset.
- Please provide more visualization similarly to those shown in Fig 4.

---

> ### Author Response · Authors · 2018-11-19
> **Thank you for the questions! We have updated our paper and answered your questions below.**
>
> Thank you for your insightful comments to help us improve our paper. First of all, we would like to mention that we add more experiments on two additional state-of-the-art strong and certified defense methods, and observe that they are also vulnerable to our proposed attacks. Please see our reply to all reviewers.
>
> Here are our responses to your concerns in “Cons” and “Minor comments”.
>
> Although we were not able to provide theoretical analysis in this paper, our proposed attacks are very effective on state-of-the-art adversarial training methods, and we believe our conclusions
> Currently, there is relatively few theoretical analysis in this field in general, and many analysis makes unpractical assumptions. We believe our results can inspire other researcher’s theoretical research.
>
> Regarding the “blind-spot attack” phrase, we are open to suggestions from the reviewers. Other phrases we considered including “evasion attack”, “generalization gap attack” and “scaling attack”. Which one do you think is a better option?
>
> Regarding the distances in Figure 3:
> Thanks for raising this concern. We have added a note to clarify this issue. The difference in distance can be partially explained by the sparsity in an adversarially trained model. As suggested in [1], the adversarially trained model by Madry et al. tends to find sparse features (see Figure 5 in [1]), where many components are zero. Thus, the distances tend to be overall smaller.
>
> Regarding the results in Table 1:
> In our old version, we only used the adversarially trained network. In our revision, we added K-L divergence computed from both adversarially trained and naturally trained networks. Additionally, we also add a new distance metric proposed by AnonReviewer1. The K-L divergences by both networks, as well as the newly added distance metric, show similar observations.
>
> Regarding adding more visualizations:
> We added some more visualizations in Fig 10 in the appendix. It is worth noting that the Linf distortion metric used in adversarial training is sometimes not a good metric to reflect visual differences. However, the test images under our proposed attack indeed have much smaller Linf distortions.
>
> We hope that we have answered all your questions, and we are glad to discuss with you if you have any further concerns about our paper.
>
> [1] Tsipras, Dimitris, et al. "Robustness may be at odds with accuracy." arXiv preprint arXiv:1805.12152 (2018).
>
> Thank you!
> Paper 1584 Authors

---

### Author Response · Authors · 2018-11-19
**Reply to All Reviewers**

During the rebuttal period, we further enhanced our experiments by conducting blind-spot attacks on two certified, state-of-the-art adversarial training methods, including (Wong & Kolter 2018) and (Singha et al. 2018). Surprisingly, although they can provably increase robustness on the training set, they still suffer from blind-spot attacks by slightly transforming the test set images. See Tables 4, and 5 in the Appendix. The attack success rates go significantly higher after a slight scale and shift on both MNIST and Fashion MNIST test sets, for both two defense models.

Additionally, we also add results for a relatively larger dataset, GTS (german traffic sign) in Appendix (Section 6.2). The results (in histograms) we observed are similar to the ones we observed on CIFAR.

With these new results, our conclusion is not limited to the adversarial training method proposed by (Madry et al. 2018). Our paper uncovers the weakness of many state-of-the-art adversarial training methods, even including those with theoretical guarantees on the training dataset. By identifying a new class of adversarial attacks, even in its simplest form (small shift + scale), many good defense methods become vulnerable again.

In conclusion, we show that many state-of-the-art strong adversarial defense methods, even including those with robustness certificates on training datasets, cannot well generalize their robustness on unseen test data from a very slightly changed domain. This partially explains the difficulty in applying adversarial training on larger datasets like CIFAR and ImageNet. We believe that our results are significant. We also think these experiments are important to further understanding adversarial examples and proposing better defenses.

---

### Author Response · Authors · 2018-11-27
**Additional small edits before the revision period closes**


We have addressed all the concerns of AnonReviewer3. During the discussion with AnonReviewer3, we found that there might be some confusions on how we generate adversarial examples from blind-spot images, and how we calculate the $\ell_p$ distortions for adversarial examples. Thus we slightly revise Section 3.3 and 4.4 to make things clear. We hope this will make our paper easier to follow.

Again we thank all the reviewers for the encouraging and constructive comments!

Thanks,
Paper1584 Authors

---

### Meta-Review · Area_Chair1 · 2018-12-14

**Confidence:** 4
**Recommendation:** Accept (Poster)

**Metareview:**

Reviewers are in a consensus and recommended to accept after engaging with the authors. Please take reviewers' comments into consideration to improve your submission for the camera ready.